# *PARK2* Microdeletion or Duplications Have Been Implicated in Different Neurological Disorders Including Early Onset Parkinson Disease

**DOI:** 10.3390/genes14030600

**Published:** 2023-02-27

**Authors:** Ausaf Ahmad, Dingani Nkosi, Mohammed A. Iqbal

**Affiliations:** Pathology and Laboratory Medicine, University of Rochester Medical Center, Rochester, NY 14642, USA; ausaf_ahmad@yahoo.com (A.A.);

**Keywords:** *PARK2*, array CGH, microdeletion, microduplication, neurodevelopmental disorder

## Abstract

The *PARK2* gene is located on 6q26, encodes ubiquitin-E3- ligase, and is a transcriptional repressor of p53. It contains 12 exons. *PARK2* copy number variants has been reported in various types of neurodevelopmental disorders, namely schizophrenia, Parkinson’s disease (PD), autism spectrum disorder (ASD), and attention-deficit/hyperactivity disorder (ADHD). In this retrospective study, nine cases (five with microdeletion and four with microduplication) are reported with 6q26 deletion disrupting the *PARK2* gene. Microdeletion sizes ranged between 215 Kb and 356 Kb, and duplication between 279 Kb and 726 Kb. These were present within the exons 7–10. Family follow up with FISH probes revealed paternal inheritance in two cases, maternal in two cases, and de novo origin in one case. Our results support previous studies showing that patients with *PARK2* CNVs involving exons 5–12 might be more deleterious and cause a unique syndrome. Comprehensive analysis of additional case studies is needed to have a full characterization of this neurological disorder syndrome.

## 1. Introduction

*PARK2* gene has ubiquitin-E3- ligase protein activity, and has been shown to be transcriptional repressor of p53 [1,2]. The Parkin protein’s main function is to regulate mitophagy and programmed cell death [3]. The schematic representation of the cDNA of the parkin gene and its molecular structure are shown in Figure 1A,B. This protein is expressed in different parts of the nervous system, such as the basal ganglia, cerebral cortex, and cerebellum [4]. Different neurodevelopmental disorders (NDD) have been associated with PARK2 including schizophrenia, early-onset Parkinson’s disease (PD), autism spectrum disorder (ASD), and attention-deficit/hyperactivity disorder (ADHD) [3,5,6,7]. The role of *PARK2* in these disorders has been extensively studied using a variety of molecular techniques. A variety of *PARK* abnormalities have been frequently implicated in NDD, namely point mutations, duplications, triplications, exonic deletions, and copy number abnormalities of different sizes. Recently, there is growing body of evidence demonstrating the significant role of *PARK2* involvement in human cancer development [8]. Different studies have shown that *PARK2*-deficient mice have an increased susceptibility to tumorigenesis and the depletion of *PARK2* in pancreatic cancer cells has been demonstrated to increase tumor formation and proliferation [8,9] 

*PARK2* pathogenicity was first implicated in early-onset PD; however, no distinct genotype–phenotype correlation was observed in patients [5]. *PARK2* has been also illustrated to be one of the rare candidate genes implicated in ASD. Many allelic variants of PARK2 has been directly linked to ASD [10]. Copy number variations (CNVs), such as chromosomal deletions or duplications, of *PARK2* have been shown to be associated with increased genetic susceptibility to ADHD [11]. Population-based analysis studies done have also, showed linkage to the *PARK2* locus 6q27 (LOD = 2.94) [12]. One of the case–control studies carried out by Glessner et al. reported exonic deletions of the *PARK2* gene in seven patients but none in the controls [6]. Other in depth analyses have also demonstrated that *VPS13B*, *WWOX*, *CNTNAP2*, *RBFOX1*, *MACROD2*, *APBA2*, *PARK2*, *GPHN*, and *RNF113A* genes play a vital role in ASD susceptibility [13]. Similar findings were also confirmed for microdeletion and microduplication encompassing *PARK2* in previous studies [7,14]. Yin et al. demonstrated that the functionality of *PARK2* can be disrupted by either exonic deletion or duplication [10]. An evidence-based review on the role of *PARK2* in ASD indicates that CNVs spanning exons 2–4 were not detrimental, while those in exons 5–12 were deleterious [15]. 

With evolving genome mapping platforms, population-based studies on ASD in the last decade have established the pathogenicity of (a) common and rare variants that are present in both patient and controls, (b) de novo and rare *PARK2* CNVs that are more common in patients, and (c) protein-disrupting single-nucleotide variants (SNVs). It is important to note that due to variable penetrance and expressivity, difficulties in genotype–phenotype correlation between proband and family members on one hand and the different cohorts on the other still exist.

We hereby report a series of cases with deletion/duplication within the 6q26 locus from the cases sent to our medical center for aCGH testing. Analysis of our data showed that all our patients’ deletions and duplications were spanning exons 7–10. This case series emphasizes what other studies have demonstrated: that *PARK2* microdeletions/microduplications spanning exons 5–10 might be more injurious, and this could be a new syndrome associated with the development of neurodevelopmental disorders [7,15].

## 2. Materials and Methods

### 2.1. Clinical Specimens

All cases with deletion/duplications within the 6q26 locus encompassing *PARK2* were identified from the aCGH laboratory dataset between 2008 and 2011. Medical records for the patients, if available, were reviewed. Otherwise, most clinical information was taken from the laboratory requisition form sent with samples.

### 2.2. aCGH Analysis

DNA was extracted using a QIAamp^®^ DNA Blood Mini Kit (Cat # 51104). Microarray experiments were performed on the Agilent 4 x44K v2.0 platform (BlueFuse Limited, Cambridge, UK). Commercially available pooled male (Cat # G147A) or female DNA (Cat # G152A) (Promega, Madison, WI, USA) was used as control DNA, i.e., patients’ DNA was referenced against same-sex control DNA. Briefly, the labeled DNA was hybridized at 65 °C for 24 h. Following washing, the slide was scanned in Agilent’s high-resolution scanner (Model #G2505C) at 3 µm resolution. The scanned image file was directly imported to BlueFuse Multi v2.5 (9271) software (BlueFuse Limited, Cambridge, UK) for the visualization and analysis of results. The log2 ratios were −0.32 for losses and 0.26 for gains.

### 2.3. FISH Analysis

FISH probes were obtained from Empire Genomics (Buffalo, NY, USA) and control FISH probes were obtained from Abbott Molecular (Abbott Laboratories, Des Plaines, IL, USA). The BAC probes were labeled either with spectrum green or spectrum orange using a nick translation kit (Cat # 07J00-001, Abbott Laboratories, IL, USA) according to the manufacturer’s instructions (See Table 1 for BAC probes and their respective controls). The probe validation was performed according to ACMG guidelines [16]. Hybridization was performed as per the manufacturer’s instructions and standard protocols. The slides were analyzed using a Nikon (Eclipse 80i) fluorescence microscope fitted with a CCD camera; Applied Spectral Imaging (ASI) software was used for image acquisition and analyses. Ten metaphases and 100 interphase cells were analyzed for the confirmation of each aCGH finding.

## 3. Results

We identified about nine patients from our database who had array CGH testing showing PARK2 gene copy number aberrations. The clinical findings and indications of all cases are shown in Table 2.

Patient #1 was a nine-year-old female with a clinical history of developmental delay, seizures, dysmorphic features, and encephalopathy. Array CGH analysis showed that she had duplications of 506 Kb on the Chr.17q21.3–17q21.32 and duplication of 347 Kb on the Chr.6q26 locus harboring the PARK2 gene (Table 2 and Table 3, Figure 2). The aberration on the Chr.17q21.3–17q21.32 region encompasses the *ARL17, LRRC37A, NSF, ARL17P1, LRRC37A, KIAA1267, LOC644246*, and *LOC51326* genes. FISH analyses of parental samples showed inheritance from the father.

Patient #2 was a one-year-old male who was referred due to Dandy–Walker syndrome and hypotonia. Array CGH analysis showed that this patient had duplication of 726 Kb on the Chr.6q26 locus harboring the *PARK2* and *PACRG* genes. FISH analyses of parental samples showed inheritance from the mother.

Patient #3 was a day-old fetus who died immediately following delivery. The autopsy results showed that cause of death was placenta abruption and cytogenetics was requested to rule out any other aberrations. Array CGH from the fetal tissue submitted showed duplication of 279 kb on the Chr.6q26 locus (Table 2 and Table 3, Figure 2).

Patient #4 was a 28-year-old female with a clinical history of developmental delays and seizures. Array CGH from the submitted sample showed duplication of 476 kb on the Chr.6q26 locus (Table 2 and Table 3, Figure 2).

Patient #5 was a five-year-old female who was referred for testing due to congenital anomaly. Array CGH analysis showed that this patient had duplication of 120 kb on the Chr. 20p13 and deletion of 215 kb on the Chr.6q26 locus (Table 2 and Table 3, Figure 2). The aberration on the Chr. 20p13 region encompasses the following genes: *DEFB125*, *DEFB126*, *DEFB127*, and *DEFB128*. FISH analyses of parental samples showed de novo origin.

Patient #6 was a nine-year-old male who had array CGH testing carried out due to autism and developmental delays. Analysis of the array CGH results showed a deletion of 346 kb on the Chr.6q26 locus (Table 2 and Table 3, Figure 2).

Patient #7 was a two-year-old male who had array testing carried out because of a clinical history of developmental delays. Array CGH results showed CNAs on the Chr.7q35, Chr.17p13.3, and Chr.19p12, and deletion of 252 kb on the Chr.6q26 locus (Table 2 and Table 3, Figure 2). The 191kb deleted region on the Chr.7q35 region harbors the *CNTNAP2* gene, and the duplication of 2.04 Mb on the Chr.17p13.3 region encompasses *PAFAH1B1*, *MDLS*, *PRPF8*, *PEDF*, *ABR*, *PITPNA*, *SMG6*, *RPA1*, *CRK*, *SRR*, and 35 more genes, while the *RPSA, ZNF681, ZNF675,* and *AK125686* genes are present in the duplicated region of 319 kb on Chr.19p12.

Patient #8 was a two-month-old female with a history of dysmorphic features and hypotonia. Deletion of 216 kb on the Chr.6q26 locus was identified using array CGH, and FISH analyses of parental samples showed the aberration was inherited from the father (Table 2 and Table 3, Figure 2).

Patient #9 was a 39-year-old male without any symptoms who had array CGH testing because of a family history of microdeletion within the 6q26 locus. His array CGH results demonstrated a 216 kb deletion on the Chr.6q26 locus (Table 2 and Table 3, Figure 2).

The schematic of chromosome 6 and all nine cases with microdeletion and microduplication of the 6q26 locus are shown in Figure 3. The microdeletion and microduplication of the first patients reported by Scheuerle et al. are also shown for comparison.

## 4. Discussion

*PARK2* microdeletions or duplications have been implicated in different neurological disorders including early onset PD and ASD [5,6,7,10]. We identified nine new cases with microdeletion/microduplication encompassing exons 7–10 of *PARK2.* At the time of testing, almost all of our patients had an associated neurological symptom present, except one patient who did not have any (patient #9). Varied clinical phenotypes have been described and associated with *PARK2* deletions/duplications including hypotonia, dysmorphic features, developmental delays, autism, and other skeletal anomalies [7,14,17]. It is reported that no distinct *PARK2* genotype–phenotype correlations exist in patients. [18]. *PARK2* is a complex multiprotein comprising of an N-terminal ubiquitin region and a highly conserved C-terminus with two RING finger domains (RING1, RING2) separated by an in-between ring (IBR) [10,19,20]. Previous publications have shown that mutations spanning exons 5–12 are more detrimental because they involve the *PARK2* coding region, which is highly conserved [15]. CNVs encompassing the C-terminus have been highly observed in patients with neurodevelopmental disorders (NDD), while CNVs targeting the N-terminus have been commonly seen in controls and cases [10,20]. *PARK2* exon 6–7 deletions are predicted to cause nonsense mutations at the 213th 213th codon of the mutant mRNA, leading to malfunction of the truncated gene because it is lacking the C-terminal domain [10].

Comparison of the clinical features present in our cohort of patients (Table 2) with the other cases published with a 6q26 deletion or duplication revealed the characteristic features associated with the syndromic neurodevelopment phenotype in the majority of the patients: developmental delay (5/9), hypotonia (2/9), and seizure (2/9). These described clinical phenotypes were not specific to either microdeletion or microduplication, suggesting that both types of aberrations within the *PARK2* gene can be involved in neurodevelopmental disorders. Furthermore, additional CNVs observed in three of our patients have been linked to a broad spectrum of neurodevelopmental disorders: duplication of 17q21.31–q21.32 (patient #1), duplication of 20p13 (patient #5) and deletion of 7q35, and duplication of 17p13.3 and 19p12 (patient #7). Microduplication of 17q21.31 has been shown to encompass the *CRHR1, MAPT*, and *KANSL1* genes, with most patients having clinical features diagnostic of ASD [21,22]. Whether these additional CNVs cause early manifestation or severe disease in association with *PARK2* remains unclear. However, such CNVs are more likely to have an effect on the different varied clinical phenotypes with which these patients present, as seen with our cohort. In PD, studies in the white population showed that there was no increased risk with having one CNV (odds ratio 1.05, *p* = 0.89) on earlier onset of PD [15]. Based on the inheritance pattern observed in our series, the parents did not have any clinical features, which supports the idea previously suggested of varied expression and incomplete penetrance of *PARK2* gene duplication or deletion [17,23].

Studies of *PARK2* in the young populations with ADHD have shown that there is a higher prevalence of *PARK2* CNVs than in controls, suggesting that copy number deletion or duplication at the *PARK2* region increases susceptibility to ADHD [11]. From our case series, patient #3 was already diagnosed with autism spectrum disorder by the time the test was requested. Follow-up on patient #4 showed that they had developmental delay, oculomotor apraxia, and ADHD by the age of nine. Comparison of our case with ASD and *PARK2* gene microdeletion to the one published by Scheuerle and Wilson showed no regions of overlap [7]. All our cases presented here with microdeletion had regions of overlap, yet no other case had been diagnosed with ASD. Additionally, cases with microduplication had regions of overlap with other reported cases of 6q26 microduplication with ADHD, but presented with varied clinical features, and only one had been diagnosed with ADHD [7,17]. Currently, it is difficult to predict which patient population, which time point, which kind of *PARK2* exonic loss or gain, or other associated factors increase the chances of developing ASD or ADHD. *PARK2* microduplications have been suggested to have more impact on the function of the gene and adverse clinical features compared to microdeletions in exons 2–4 of the *PARK2* gene [10]. The frequency of *PARK2* deletions between exons 2 and 3 has been reported to be same in patients and controls, suggesting that these are not a major risk factor for ASD [13]. However, our data show that there is a subset of individuals with *PARK2* CNVs involving exonic deletion on 7–10 who do not develop any neurological associated symptoms, which might be due to incomplete penetrance or dosage. Taken together, these results show that more studies of individuals carrying *PARK2* CNVs are required so that a comprehensive characterization can be carried out to correlate clinical features and associated 6q26 microdeletion/microduplication regions to further describe this newly suggested syndrome.

The varied age of presentation and the different clinical features of the patients, with all harboring CNVs within the 6q26 locus, necessitate that we might have to perform proper neurological examinations in order to fully characterize genotype–phenotype correlation with 6q26 microdeletion/microduplication syndrome. To our knowledge, the minimum deletion or duplication size required and which exonic involvement leads to the development of the neurodevelopmental disorders is yet to be established. Mutations involving exons 2–9 are frequent in PD. However, in our cohort, exons 7–10 were observed with higher percentage. This reinforce the hypothesis that *PARK2* CNVs might be more damaging, and this is more evident in patients with exonic duplications. Additional similar case studies will provide more evidence on the role of *PARK2* in PD.

## Figures and Tables

**Figure 1 genes-14-00600-f001:**
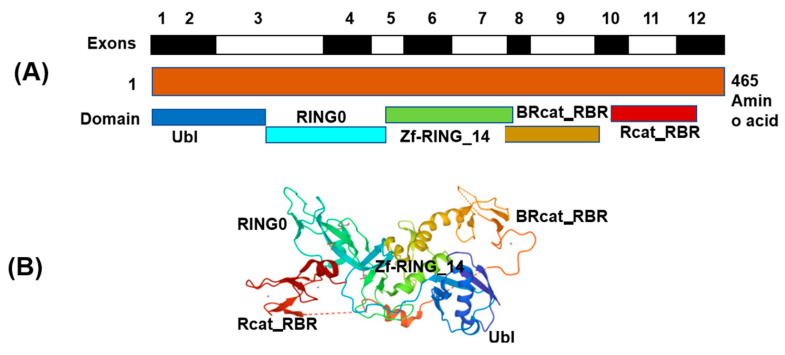
(**A**) Schematic presentation of cDNA of the parkin (PARK2) gene exons and (**B**) Structure of full-length Parkin (PDB 5C1Z). Note: The protein structure was downloaded from RCSB PDB: Homepage and Cn3D macromolecular structure viewer (Cn3D Home Page (nih.gov) was used for preparation of schematic figure. The presentation is not to the scale.

**Figure 2 genes-14-00600-f002:**
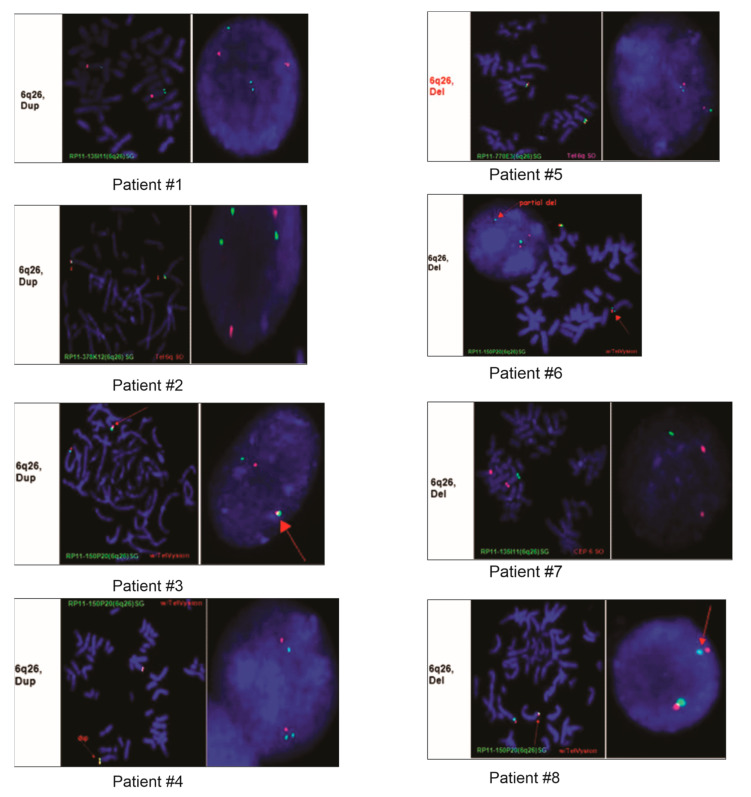
FISH images from eight of the nine patients demonstrating duplication or deletion within the 6q26 region.

**Figure 3 genes-14-00600-f003:**
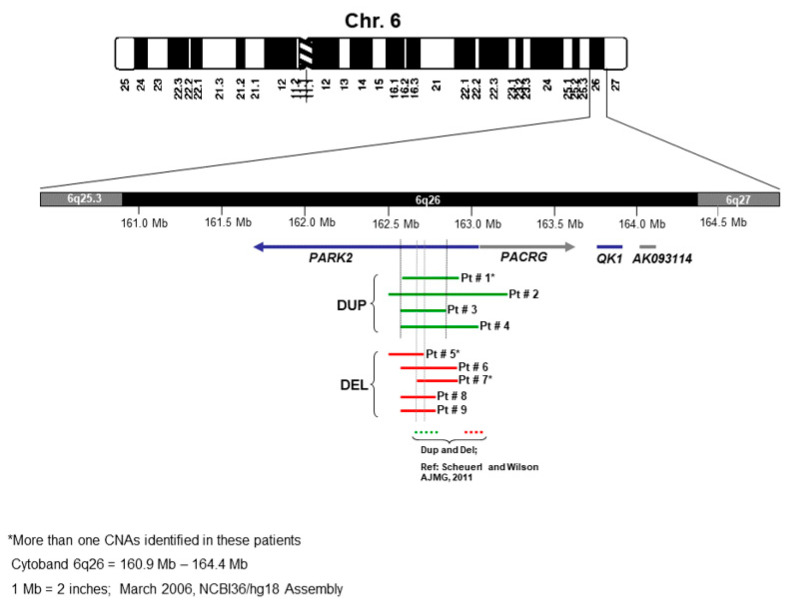
Schematic of chromosome 6 and comparison of the nine cases with deletion/duplication of 6q26 locus (see Table 2).

**Table 1 genes-14-00600-t001:** Summary of BAC and control probes for validation of aCGH findings.

Patient ID	BAC Probe (Test)	Control Probe
Pt. # 1	RP11-135I11, Spectrum Green (6q26) and RP11-609G9, Spectrum Green (17q21.31–q21.32)	TelVysion 6q, Spectrum Orange TelVysion 17p, Spectrum Orange
Pt. # 2	RP11-378K12, Spectrum Green	TelVysion 6q, Spectrum Orange
Pt. # 3	RP11-150P20, Spectrum Green	W/TelVysion, Spectrum Orange
Pt. # 4	RP11-150P20, Spectrum Green	W/TelVysion, Spectrum Orange
Pt. # 5	RP11-770E3, Spectrum Green(6q26) and TelVysion 20p/20q, Spectrum Green (20p13)	TelVysion 6q, Spectrum Orange and TelVysion 20q, Spectrum Orange
Pt. # 6	RP11-135I11	
Pt. # 7	RP11-135I11, Spectrum Green (6q26); RP11-64O16, Spectrum Green (7q35); RP11-818O24, Spectrum Orange (17p13.3); and RP11-48K6, Spectrum Orange (19p12)	CEP 6, Spectrum Orange; TelVysion 7q, Spectrum Orange; TelVysion 17p, Spectrum Green; and TelVysion 19p, Spectrum Green
Pt. # 8	RP11-150P20, Spectrum Green	W/TelVysion, Spectrum Orange
Pt. # 9	RP11-150P20, Spectrum Green	W/TelVysion, Spectrum Orange

**Table 2 genes-14-00600-t002:** Summary of the clinical findings, indications, and PARK2 gene status. Case 1–4 with duplication and case 5–9 with deletion.

Clinical Features/Patient	1	2	3	4	5	6	7	8	9
*PARK2* status	**Duplication**				**Deletion**				
Developmental delay	Yes	Yes		Yes		Yes	Yes		
Seizures	Yes			Yes					
Dysmorphic features	Yes							Yes	
Hypotonia		Yes						Yes	
Congenital anomalies		Yes	Yes		Yes				
Autism spectrum disorder						Yes			
ADHD		Yes							
Encephalopathy	Yes								
Size of deletion or duplication	347 Kb	726 Kb	279 Kb	476 Kb	215 Kb	346 Kb	252 kb	216 Kb	216 Kb

**Table 3 genes-14-00600-t003:** Summary of the aCGH results of the nine cases.

Case	Age	Sex	Cytoband	Coordinates	Size	Del/Dup	Exons Involved	Origin	Known Genes
Pt # 1	9 Y	F	6q26 and 17q21.31–q21.32	162573881_16292045141544024_42049740	347 kb 506 kb	Dup Dup	7–9	Father	**6q26** = *PARK2* **17q21.31–q21.32** = *NSF*, *ARL17P1*, *LRRC37A*, *KIAA1267*, *LOC644246*, *LOC51326*
Pt # 2	1 Y	M	6q26	162504320_163230798	726 kb	Dup	7–12	Mother	*PARK2 and PACRG*
Pt # 3	33 Y 7 M	F	6q26	162574111_162853117	279 kb	Dup	7–8	NA	*PARK2*
Pt # 4	28 Y 9 M	F	6q26	162574111_163049684	476 kb	Dup	7–10	NA	PARK2
Pt # 5	5 Y 3 M	F	6q26 and 20p13	162504090_162719512 18380_138265	215 kb 120 kb	Del Dup	7	de novo	**6q26** = *PARK2* **20p13** = *DEFB125*, *DEFB126*, *DEFB127*, *and DEFB128*
Pt # 6	9 Y 3 M	M	6q26	162574111_162920281	346 kb	Del	7–9	NA	PARK2
Pt # 7	2 Y 6 M	M	6q26; 7q35; 17p13.3; and 19p12	162668308_162920281 145998312_146189705 915669_2952347 23604133_23924885	252kb 191 kb 2.04 Mb 321 kb	Del Del Dup Dup	8–9	NA	**6q26** = *PARK2* **7q35** = *CNTNAP2* **17p13.3** = *PAFAH1B1*, *MDLS*, *PRPF8*, *PEDF* **19p12** = *RPSA*, *ZNF681*, *ZNF675, AK125686*, *BC015383*, *LOC370087, BC082233*
Pt # 8	2 M	F	6q26	162574111_162790389	216 kb	Del	7	Father	*PARK2*
Pt # 9	39 Y 2 M	M	6q26	162574111_162790389	216 kb	Del	7	NA	*PARK2*

## Data Availability

The authors declare that all data supporting the findings of this study are available within the article and its supplementary information files or from the authors upon request.

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
