# Peer review of "PARK2 Microdeletion or Duplications Have Been Implicated in Different Neurological Disorders Including Early Onset Parkinson Disease"

_genes, 2023, doi:10.3390/genes14030600_

Round 1

Reviewer 1 Report

The aim of this paper is to present an additional 9 cases of duplications or deletions involving the PARK2 gene uncovered by a clinical laboratory using array CGH methods. These dup/dels were all within exons 7-10 of the 12 exons in PARK2. The methods used and findings are generally well described. The discussion should include some discussion of the known functions of this exonic region.

Specific comments

Either Parkinson (Line 30) or Parkinson’s (L 232) should be consistently used throughout.

Numbered references are missing in several locations (L59), (L97).

Table 2 should be on one page and the PARK2 status line is confusing. It should be deleted and the last line Size of Del/Dup should be made into two lines to clarify which they are.

A description for patient 4 is missing (L132), and patient 9 should be moved to the bottom, rather than before patient 5.

The beginning sentence (L172) of the discussion is too repetitive of the introduction and should be deleted.

A word (be) is missing in line 229 that should be before done.

The sentence beginning on Line 234 might be better off to use the word allow rather than make.

It would be of interest to correlated the speculation with some discussion of the role these exons might contribute to the structure and function of PARK2 [This could be related previous research, such as Beasley, PNAS,2007, v40, p3095]. Or others, such as a figure from Wikipedia

Author Response

The authors sincerely thanks the reviewer for making comments on our manuscript.

General Comments:

Response: Known functions of the PARK2 exons is included in the discussion... in L213-221 and L266-271

Specific comments

Either Parkinson (Line 30) or Parkinson’s (L 232) should be consistently used throughout.

Response: Parkinson is used throughout the manuscript

Numbered references are missing in several locations (L59), (L97).

Response: References added L79….. associated with the development of neurodevelopmental disorders[4,15]

L118-119: The probe validation was performed according to ACMG guidelines [16].

Table 2 should be on one page and the PARK2 status line is confusing. It should be deleted and the last line Size of Del/Dup should be made into two lines to clarify which they are.

Response: Revised according to the reviewer comment

A description for patient 4 is missing (L132), and patient 9 should be moved to the bottom, rather than before patient 5.

Response: Patient 4 is added …L155-157.

Patient 9 is moved to bottom.. L179-181

The beginning sentence (L172) of the discussion is too repetitive of the introduction and should be deleted.

Response: The sentence is modified to remove repetitiveness and the authors feel that it is important to mention it for the sake of discussion.L203-205

A word (be) is missing in line 229 that should be before done.

Response: The sentence is revised and corrected ….L279-283

The sentence beginning on Line 234 might be better off to use the word allow rather than make.

Response: Edited….L287-290

It would be of interest to correlated the speculation with some discussion of the role these exons might contribute to the structure and function of PARK2 [This could be related previous research, such as Beasley, PNAS,2007, v40, p3095]. Or others, such as a figure from Wikipedia

Response: The role of PARK2 exons is elaborated in L213-221 and L266-271

Reviewer 2 Report

The reviewer's report is attached.

Author Response

The authors sincerely thank the reviewer for their comments.

1. The microdeletions and microduplications identified in this study are new and would be relevant if these changes alter the function of the PARK2 protein. If that is beyond the scope of this paper, then at least whether these variations alter the mRNA expression of PARK2 gene or not, that needs to be answered.

Response: The authors sincerely thank the reviewer for the  comments. The function of PARK2 exons is included in the discussion ..L213-221

  1. The description of patient #8 and patient #9 (line no. 128-132) is not matching with Table 2 and Table 3. The authors should correct this (I am assuming these would be patient #3 and patient #4 respectively).

Response: Thank you for pointing this error and it is fixed in Tabe 2 and 3

Patient #8 was two months female with history of dysmorphic features and hypotonia. Deletion of 216 kb on Chr.6q26 locus was identified by array CGH and FISH analyses of parental samples showed aberration was inherited from the father (Table 2, 3, Figure 1).

Patient #9 was a 39-year male without any symptoms and had array CGH testing because of family history of microdeletion within 6q26 locus His array CGH results demonstrated a 216 kb deletion on Chr.6q26 locus (Table 2, 3, Figure 1)

3. Reference no. is missing in line 59.

Response: The reference is added…L59

4. There are few grammatical errors that need to be corrected.

Response: Thanks for the comment. The manuscript is revised to fix the grammatical mistakes.

Round 2

Reviewer 2 Report

The authors addressed all the comments and changed the manuscript accordingly. The revised manuscript is significantly better and I recommend that this can be accepted for publication.